# Failure Analysis and Accelerated Test Development for Rotor Magnetic Bridge of Electric Vehicle Drive Motor

**Sibo Wang** [1,2], **Jing Shang** [1,*], **Lihui Zhao** [3], **Le Li** [3], **Zhen Wang** [3], **Dazhi Wang** [2] **and Xiaoxu Wang** [2]

1   Department of Electric Engineering, Harbin Institute of Technology, Harbin 150006, China; wangsibo@faw.com.cn
2   China FAW Group Co., Ltd., Changchun 130013, China; wangdazhi@faw.com.cn (D.W.); wangxiaoxu@faw.com.cn (X.W.)
3   School of Mechanical Engineering, University of Shanghai for Science and Technology, Shanghai 200093, China; lihui_zhao@usst.edu.cn (L.Z.); power_lile@126.com (L.L.); wangzhenares@yeah.net (Z.W.)
*   Correspondence: shangjing@hit.edu.cn

**Abstract:** Motor rotor magnetic bridges operate under multiple physical field loads, such as electromagnetic force, temperature, and centrifugal force. These loads can cause fatigue and aging failure of the bridges, especially when the rotor is operating continuously at high speeds and high temperatures. Therefore, the failure analysis and accelerated test cycle development of magnetic bridges is a major aspect of their reliability evaluation. This paper studies rotor multi-physics load transfer characteristics and establishes a rotor magnetic bridge failure physical model. A simulation analysis is conducted from the electromagnetic field, thermal field, structural field, and thermomechanical coupling field to determine the risk point load responses and failure-dominant loads. In addition, the accuracy of the simulation model is verified by actual bench tests. Considering the influence on the rotor bridge's life under the coupling of multiple failure modes, the fatigue failure model under alternating loads and the fatigue aging coupling failure model are established, respectively. Through a damage analysis, the whole life cycle damage targets for both failure modes are determined, and the test condition levels are screened based on the load frequency distribution and damage distribution. The multi-objective optimization method is used to calculate the number of test cycles and finally develop accelerated test cycle conditions that can reproduce multiple failure modes. This research can provide support for rotor bridge reliability design and verification, as well as product quality development.

**Keywords:** rotor magnetic bridge; multi-physical fields; failure modes; reliability test; load spectrum development



## 1. Introduction

Currently, the electric drive system, as the core component of the electrification of new energy vehicles, is developing towards high power density, high efficiency, high reliability, and safety, among other characteristics [1]. Due to their efficient performance, permanent magnet synchronous motors are widely used in electric drive systems. As a critical component in permanent magnet synchronous motors, the rotor is subjected to multiple physical loads, such as electromagnetic force, thermal stress, and centrifugal force, during its operation. There are numerous failure modes, such as magnetic bridge fatigue [2], winding aging [3,4], and permanent magnet demagnetization [5,6]. In addition, the risk of rotor magnetic bridge failure is further exacerbated by the increased intensity of the actual complex alternating loads. It is essential to consider the failure of rotor magnetic bridges under multiple physical fields and to build a magnetic bridge accelerated test load spectrum that reflects actual user conditions to verify product reliability levels and support high-quality development.

In recent years, many scholars have researched the failure load and reliability of the permanent magnet synchronous motor rotor. Wu et al. [7] used a finite element simulation to analyze the stress distribution in the rotor under rated speed conditions. They concluded that the centrifugal force on the rotor structure is proportional to the square of the speed. The higher the speed, the greater the centrifugal force. This method only considers the centrifugal forces on the magnetic bridge under alternating speeds. However, they ignore the effect on the life of the magnetic bridge due to thermal stresses caused by alternating temperatures, such as thermal conduction and thermal concomitant, as well as the electromagnetic forces generated by torque fluctuations during actual operation. Liu et al. [8] pointed out that the external forces on the rotor during the process mainly include centrifugal force, electromagnetic force, and thermal stress caused by the temperature rise in the rotor. The simulation analysis shows that the electromagnetic forces are small compared to the centrifugal forces and thermal stresses and can be neglected at high motor speeds. However, they only considered the steady-state condition in the simulation analysis process. Meanwhile, the load on the rotor is complex and dynamic in the actual process, and a transient condition analysis should be further considered. In addition, the damage contribution to the magnetic bridges under different stresses can be further analyzed in terms of damage or lifetime in conjunction with their physical failure model when defining the proportion of different physical loads. Eerik Sikanen et al. [9,10] evaluated the lifetime of a motor rotor based on transient operating conditions, taking into account electromagnetic-thermomechanical stresses. However, they only studied the rotor structure of the in-line design, and the conclusion of the dominant failure load was not convincing. Most importantly, the effect of temperature on the S–N curve of the material was ignored when the lifetime assessment was performed.

In addition, the rotor magnetic bridge is influenced by multiple sources of load and can experience various failure modes, such as fatigue and aging. Currently, scholars have conducted many studies on the development of accelerated test load profiles [11,12], which mainly involve load data acquisition, condition construction, target definition, and cycle number determination. Lihui Zhao et al. [1] studied the construction of accelerated test cycle conditions for mechanical components of electric drive systems based on user data and mainly considered fatigue failure. Jinhao Cai et al. [13] analyzed the failure mechanism of the power device IGBT and accelerated the assessment of its thermal aging failure through power cycle conditions. At present, existing motor reliability test standards conduct accelerated tests under rated and peak conditions, such as GB/T 29,307 and ISO 19,453.4, but they do not consider the coupling of multiple failure modes. This approach is insufficient to reproduce the simultaneous occurrence of multiple failure modes of rotor bridges in the actual operation of users. Developing accelerated test load cycles that cover multiple failure modes and consider lifetime targets during actual user operation is crucial for product reliability evaluation.

Therefore, this paper establishes a physical model of magnetic bridge failure based on the actual physical model of the magnetic bridge and its operating conditions. The paper carries out a simulation analysis from electromagnetic, thermal, and thermomechanical coupling fields to determine the location of the rotor danger point. By analyzing the magnetic bridge damage contribution under multi-physical field loading, the dominant failure loads and failure modes are identified.

Fatigue tests on the rotor materials are carried out, and life models for the magnetic bridges are established at different temperatures and under alternating speed conditions. Reliability test cycles are constructed from user data to match the actual user loads and determine the damage targets, which contain fatigue aging coupled damage and alternating loads fatigue damage. Finally, an accelerated reliability test load spectrum is developed. The research presented in this paper offers a clear and reliable approach for compiling acceleration test load spectra, which can effectively support the durability assessment of electric motor rotors. This approach is beneficial for improving the efficiency of product

development and validation, thereby providing support for the high-quality development of electric motor rotors. Figure 1 illustrates the main technical route in this paper.

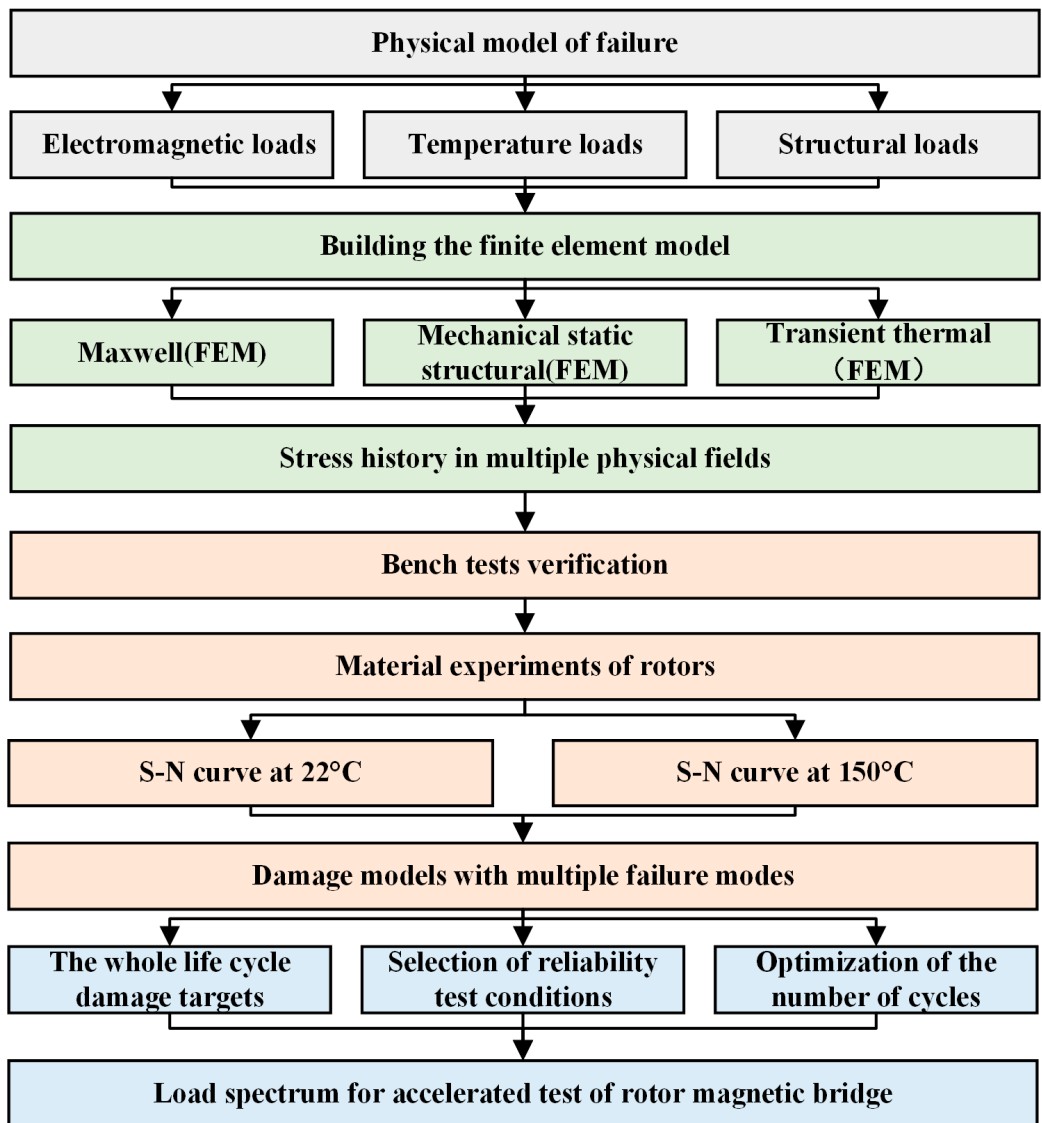

**Figure 1.** Technical routes.

## 2. Physical Model of Failure

### 2.1. Analytical Model for Electromagnetic Loads

Electric vehicle motor rotors are typically made of stacked electrical steel sheets that are less than 1 mm thick. In addition to meeting primary electromagnetic performance conditions, the rotor must withstand substantial centrifugal force at extremely high speeds, thermal stress under varying temperatures, and electromagnetic forces. The dynamic alternating stress response can cause fatigue failure of the rotor magnetic bridge, resulting in motor failure. Therefore, analyzing the specific causes of motor rotor failure requires mathematical models in the electromagnetic, thermal, and structural fields to obtain the stress distribution in the rotor under each physical area.

The rotor rotates due to the electromagnetic force generated by the interaction of the magnetic field and the current. The calculation equation for electromagnetic stress is as follows [14]:

$$\left. \begin{array}{l} \sigma_{ij,j} f_i = 0, \\ \varepsilon_{ij} = \frac{1}{2}\left(u_{i,j} + u_{j,i}\right), \\ \varepsilon_{ij} = \frac{1+v}{E}\sigma_{ij} - \frac{v}{E}\sigma_{kk}\delta_i. \end{array} \right\} \tag{1}$$

where $\sigma_{ij}$ is the stress on the rotor; $f_i$ is the Lorentz force; $\varepsilon_{ij}$ is the strain on the rotor; $\sigma_{ij,j}$ is the derivative of the stress to the displacement; $u$ is the displacement; $v$ is the Poisson's ratio; $E$ is the elastic modulus; $\varepsilon_{ij}$ is a factor of the equation; and n is 1 when $i = j$ and 0 when $i \neq j$.

### 2.2. Analytical Model for Temperature Loads

Motor components generate various losses in the operation process, mainly including stator and rotor core losses, eddy current losses of permanent magnets, copper losses of windings, and mechanical losses. Losses are the source of heat that causes temperature variations in various parts of the motor. Therefore, when solving for the temperature distribution of the motor, the heating rate of each part of the motor must be obtained. The core losses of the stator and rotor as well as the eddy current losses of the permanent magnets can be obtained by an electromagnetic simulation. Assuming the speed of the motor is 1000 rpm and since eddy current losses are proportional to the square of the frequency, Equation (2) can be used to derive the losses for other operating conditions at 1000 rpm, thus covering the entire operating condition at the current speed [9].

$$P(n, T) = \left(\frac{n}{1000}\right)^2 P(1000, T) \tag{2}$$

where $P(n, T)$ is the eddy current loss when the speed is $n$ rpm, and the torque is $T$. The winding copper consumption and mechanical losses can be calculated using the empirical formulae in the literature [15]. The calculation of the motor thermal field is based on the basic theory of thermodynamics. In cartesian coordinates, the boundary conditions and calculation equations of the three-dimensional heat conduction problem are as follows [16]:

$$\left. \begin{array}{l} \frac{\partial}{\partial x}\left(\lambda_x \frac{\partial T}{\partial x}\right) + \frac{\partial}{\partial y}\left(\lambda_y \frac{\partial T}{\partial y}\right) + \frac{\partial}{\partial z}\left(\lambda_z \frac{\partial T}{\partial z}\right) + qv = \rho c \frac{\partial T}{\partial t}, \\ -\lambda \frac{\partial T}{\partial n}\Big|_{s1} = q_0, \\ \lambda \frac{\partial T}{\partial n}\Big|_{s2} = -\alpha(T - T_w). \end{array} \right\} \tag{3}$$

where $\lambda_x$, $\lambda_y$, and $\lambda_z$ are the thermal conductivity of the object in the $x$, $y$, and $z$ directions in the thermal field; $T$ is the motor temperature; $qv$ is the heat source density of each part inside the motor; $\rho$ is the density; $c$ is the specific heat capacity; $t$ is the time; $s_1, s_2$ is the edge surface of the thermal field; $q_0$ is the heat flux density across the boundary surface $s_1$; $n$ is the boundary vector of the object; $T_w$ is the temperature of the surrounding objects at the boundary $s_2$; $\alpha$ is the heat dissipation coefficient of the edge surface. The stator surface dissipation coefficient $\alpha_s$ and the rotor surface heat dissipation coefficient $\alpha_r$ can be calculated using Equation (4) [17].

$$\left. \begin{array}{l} \alpha_s = \frac{1+0.04v}{0.045} \\ \alpha_r = 28(1 + \sqrt{0.45v}) \end{array} \right\} \tag{4}$$

where $v$ is the circumferential linear speed of the rotor. When the internal temperature of the motor changes, a temperature gradient is formed within the rotor due to the different heat generation rates in the structure, and thermal stresses are generated as a result [18]. Based on the theory of thermo-elasticity, the equation for calculating the thermal stress on the structure is as follows [19].

$$\left. \begin{array}{l} \sigma_{ij,j} = 0, \\ \varepsilon_{ij} = \frac{1}{2}\left(u_{i,j} + u_{j,i}\right), \\ \varepsilon_{ij} = \frac{1+v}{E}\sigma_{ij} - \frac{v}{E}\sigma_{ki}\delta_{ij} + \beta\Delta T. \end{array} \right\} \tag{5}$$

where $\beta$ is the coefficient of thermal expansion of the rotor and $\Delta T$ is the temperature variation.

### 2.3. Analytical Model for Structural Loads

The high-frequency alternating torque and speed loads subject the motor rotor to high mechanical stresses. The method of transient dynamics is generally used to calculate the mechanical stresses, as shown in the following equation [20].

$$\left.\begin{array}{l} \sigma_{ij,j} + f_i = \rho \frac{\partial^2 u_i}{\partial t^2}, \\ f_i = m\omega^2 r, \\ \frac{1}{2}\left(u_{i,j} + u_{j,i}\right) = \frac{1+v}{E}\sigma_{ij} - \frac{v}{E}\sigma_{kk}\delta_{ij}. \end{array}\right\} \tag{6}$$

where $\rho$ is the density of the rotor structure, $f_i$ is the centrifugal force, $m$ is the mass of the rotor structure, $\omega$ is the angular velocity of the rotor, and $r$ is the radius of the rotor.

## 3. Analysis of Failure Modes in Multiple Physical Fields

### 3.1. Models and Parameters

A cross-sectional view of a PMSM for electric vehicles studied in this paper is shown in Figure 2. The motor is an eight-pole 48-slot V-shaped rotor PMSM, which consists of four parts: the stator, winding, rotor, and permanent magnet. Its main parameters are shown in Tables 1 and 2.

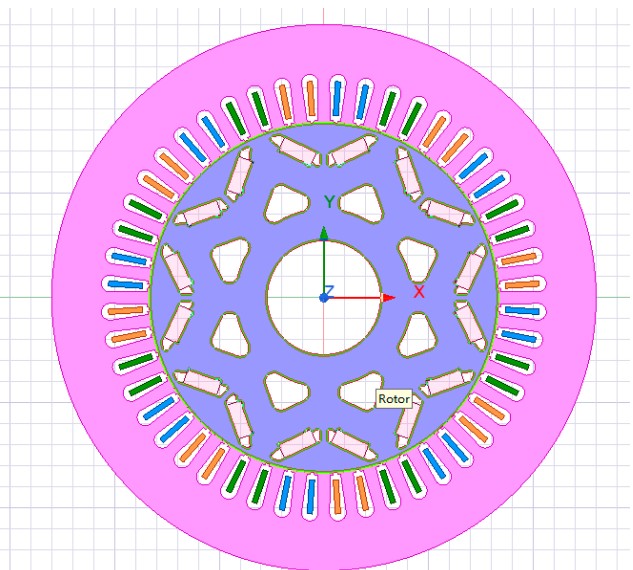

**Figure 2.** 2D diagram of motor.

**Table 1.** Dimensional parameters of PMSM.

| Parameters | Value/mm |
| --- | --- |
| Stator outer diameter | 260 |
| Stator bore diameter | 166.8 |
| Stator slot height | 21.6 |
| Stator lamination thickness | 160 |
| Rotor lamination thickness | 160 |
| Length of permanent magnets | 17 |
| Width of permanent magnets | 7.4 |

This paper simulates the operation of a motor on a bench test through a finite element analysis (FEA) based on the actual acquisition of motor operation data. The results obtained from the simulation are compared with the load data monitored by the test to demonstrate the accuracy of the simulation model. The dynamic alternating test condition data are shown in Figure 3, with a maximum motor speed of 8907 rpm and a maximum current of 200 A.

**Table 2.** Material parameters of PMSM.

| Parameters | Permanent Magnets | Stator and Rotor | Adhesive |
|---|---|---|---|
| Density (kg/m$^2$) | 7600 | 7650 | 1207 |
| Young's modulus (MPa) | 150,000 | 200,000 | - |
| Poisson's ratio | 0.24 | 0.3 | - |
| Yield strength (MPa) | 75 | 470 | - |
| Ultimate strength (MPa) | 290 | 585 | - |
| Specific heat (J/(kg C)) | 450 | 460 | 1173 |
| Thermal expansion (1/K) | $6 \times 10^{-6}$ | $1.2 \times 10^{-5}$ | - |
| Thermal Conductivity (W/(mK)) | 8.5 | 39 | 0.3 |
| Conductivity (siemens/s) | $5.6 \times 10^5$ | $2 \times 10^6$ | - |

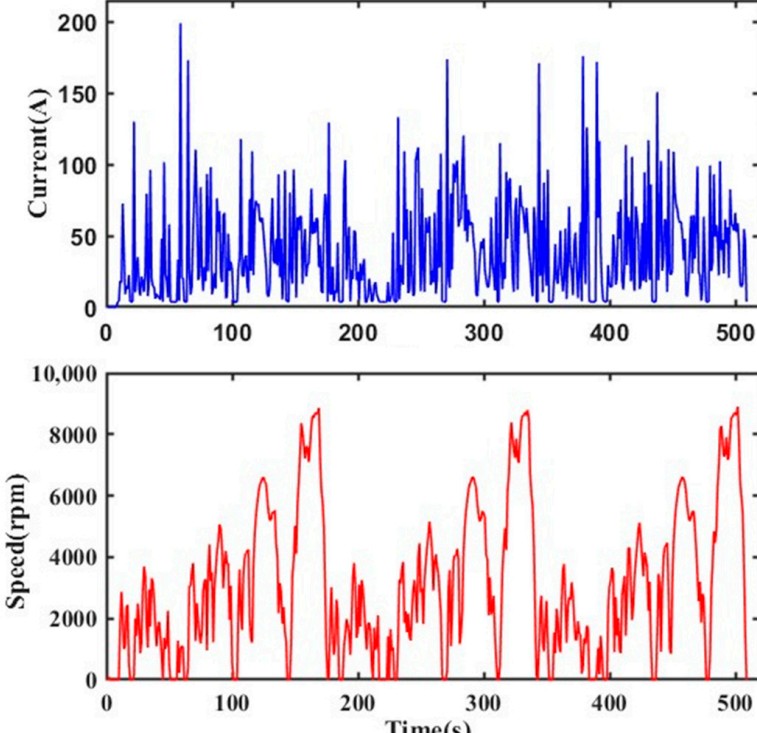

**Figure 3.** Current and speed time history.

### 3.2. Electromagnetic Field Load Analysis

Since the stator, rotor, and permanent magnets are evenly distributed along the axial direction of the motor, a two-dimensional model can be used in the electromagnetic finite element simulation to ensure calculation accuracy. During the electromagnetic simulation, it was not possible to replicate the transient conditions shown in Figure 3. Therefore, we selected the most extreme conditions, such as a current of 200 A and a speed of 8907 rpm, for the simulation to verify the damaging effect of electromagnetic forces. By taking advantage of the symmetry and periodicity of the overall motor, we simplified the motor model to one-eighth scale for our analysis, which reduced the computational workload.

Figures 4 and 5 show the electromagnetic force density and the electromagnetic force waveform on the rotor surface of the motor under extreme operating conditions, respectively. The rotor's surface electromagnetic force density is represented by a short and dense blue arrow, with a maximum force of 52 N.m. The red arrow represents the electromagnetic force density at the winding coil. We extracted the electromagnetic force distribution on the rotor structure and imported the results into the structure field to analyze the stress distribution on the rotor structure. Figure 6 is a cloud diagram showing the stress distribution under the action of electromagnetic force alone. The maximum stress

point of the rotor occurs at the junction of the permanent magnet and the rotor slot, with a magnitude of 2.49 MPa. Meanwhile, the stress at the magnetic isolation bridge is only 1.5 MPa. Therefore, the effect of the electromagnetic force can be ignored.

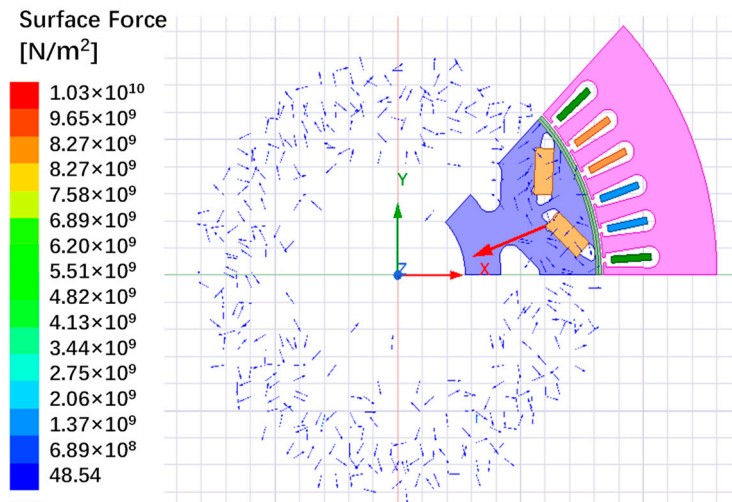

**Figure 4.** Electromagnetic force density distribution diagram.

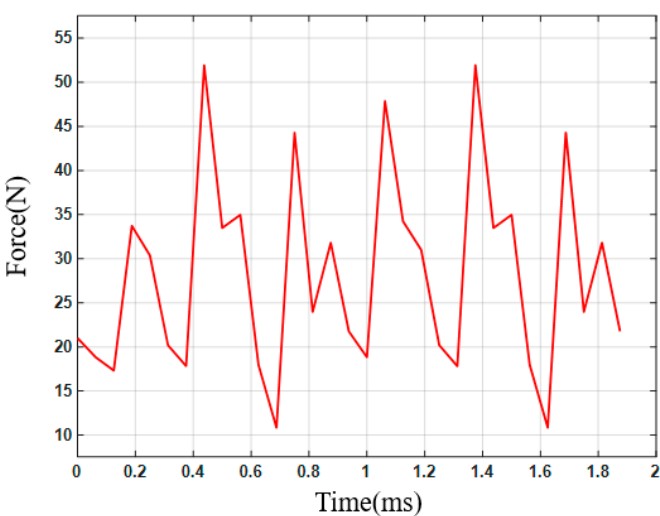

**Figure 5.** Electromagnetic force time history.

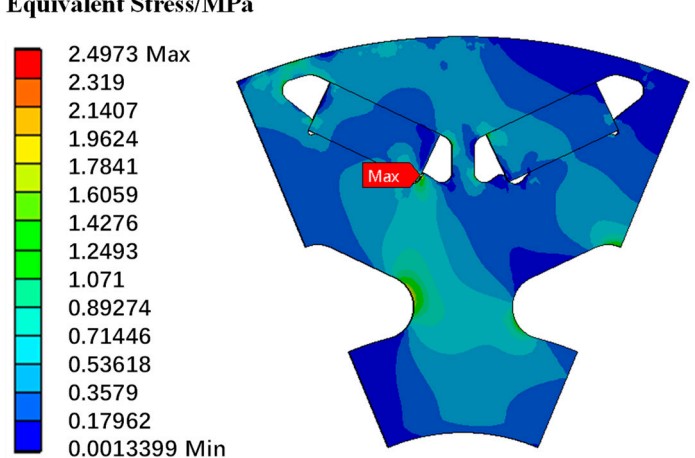

**Figure 6.** Electromagnetic stress distribution cloud map.

### 3.3. Thermal Field Load Simulation and Verification

In the thermal field simulation process, the loss values calculated under the electromagnetic field are first imported into the thermal field as a heat source, and the parameters are set, such as the thermal conductivity and specific heat capacity of each motor component. The finite element model is carried out by setting the boundary condition of the heat dissipation coefficient on the outer surface of the stator rotor structure.

As shown in Figures 7 and 8, the motor produced a temperature rise in all parts when operating under the test conditions, with the highest temperature of 87.481 °C occurring in the winding section. This is due to the high Joule heat generated by the three AC currents passing through the winding coil and the poor heat dissipation conditions around the coil. The stator core temperature is higher than that of the rotor area because the losses in the stator core are much higher than in the rotor core.

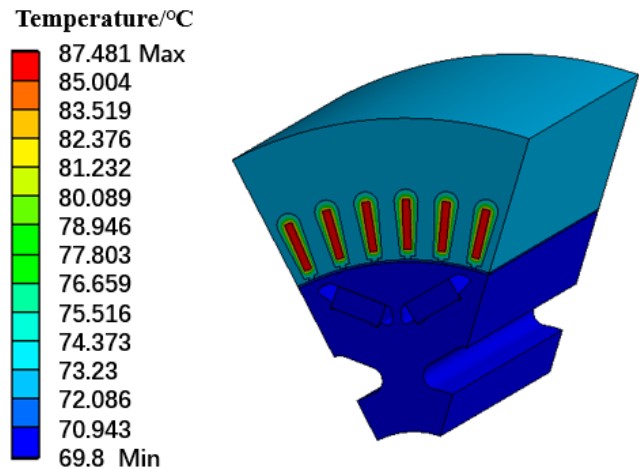

**Figure 7.** Motor temperature distribution diagram.

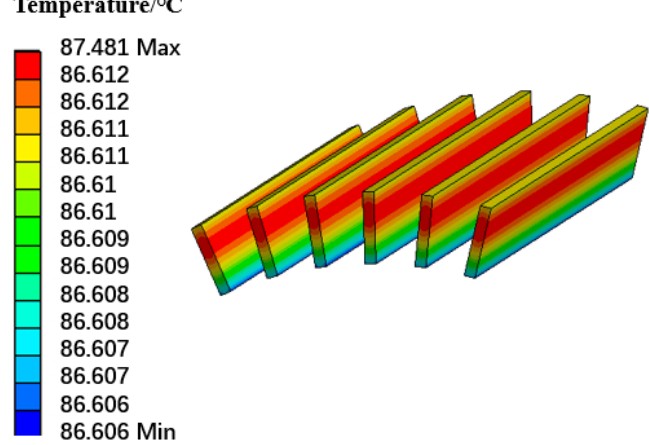

**Figure 8.** Stator winding temperature distribution diagram.

As shown in Figure 9, the highest temperature of the rotor occurs near the magnets, with a maximum temperature of 70.405 °C. Due to the relatively large eddy current losses in the permanent magnets, the rotor temperature decreases from the inside to the outside along the permanent magnet area, while the rotor core losses are mainly concentrated near the magnets. The temperature of the outer surface of the rotor is lower than the temperature of the entire rotor. With the change in rotation speed, the outer surface of the rotor will have a better air convection environment to dissipate heat, indicating that the temperature rise of the stator has little effect on the rotor due to the air gap between the stator and the rotor. Figure 10 shows the temperature change curve at the magnetic bridge.

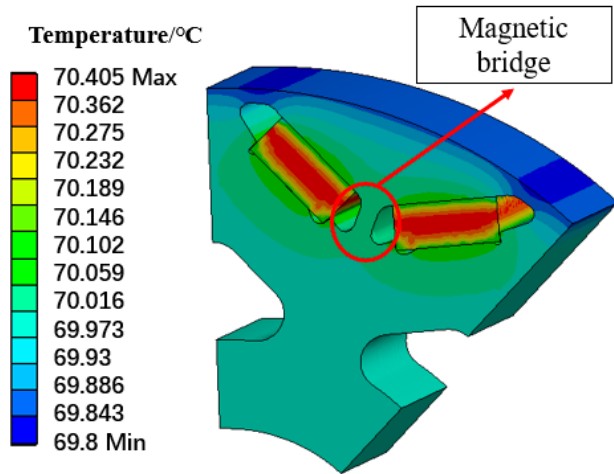

**Figure 9.** Rotor temperature distribution diagram.

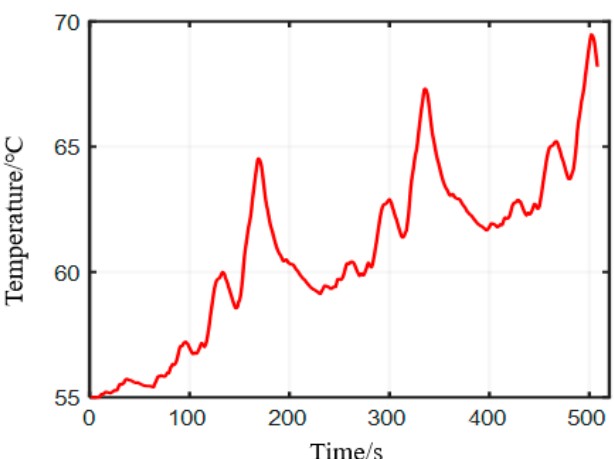

**Figure 10.** Temperature variation at the rotor spacer bridge.

Temperature sensors were installed on the stator during testing. However, it was challenging to install sensors on the rotor's rotating parts, and as a result, the rotor temperature followed the same trend as the winding temperature. To verify the accuracy of the simulation model established in this paper, the winding temperature data were monitored and compared with the simulated stator winding temperature during the bench test, as shown in Figure 11.

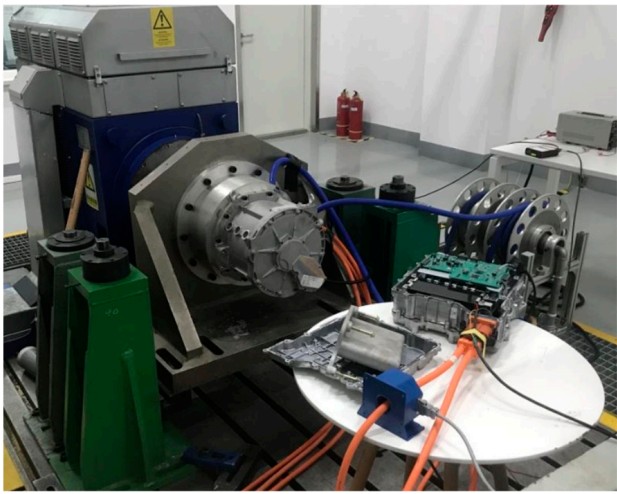

**Figure 11.** Bench test diagram.

Figure 12 displays a comparison of the measured stator temperature with the simulated temperature. The figure indicates that the simulated temperature and the measured temperature follow the same trend in the time domain, but with a certain error. This is due to simplifying the number of stator winding meshes and omitting the winding ends in the simulation process, as well as using an empirical formula for the boundary heat dissipation coefficient of the stator and rotor. The error between the simulated combination and the actual temperature is within 5%, which is acceptable and verifies the accuracy of the thermal field simulation model.

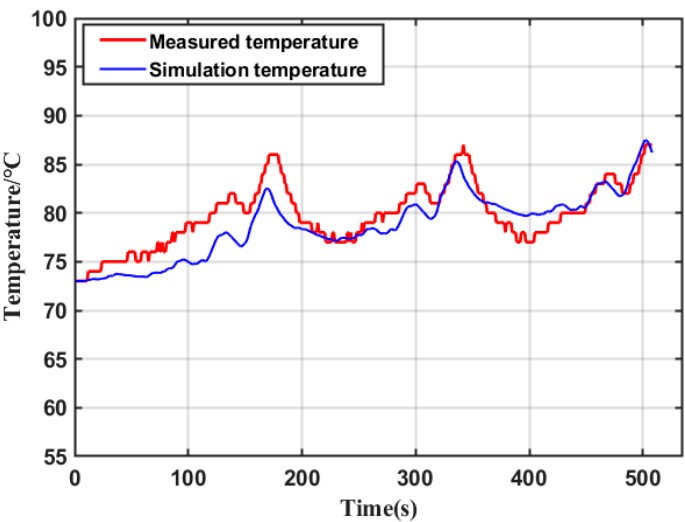

**Figure 12.** Comparison of simulated and measured temperatures.

### 3.4. Structural Field Load Analysis

During high-speed operation, the rotor is mainly subjected to centrifugal forces. The dynamic alternating conditions during vehicle driving cause the motor rotor to generate large, time-varying centrifugal forces. As the speed increases, the centrifugal force on the rotor also increases.

For the structural field analysis, a non-separating connection was set between the permanent magnet and the rotor core to allow for slight slippage while maintaining the closure between the permanent magnet and the rotor core. When setting the cylindrical symmetry constraints, radial and axial displacements were allowed, while tangential displacements were constrained. The specific simulation results are shown in Figures 13 and 14.

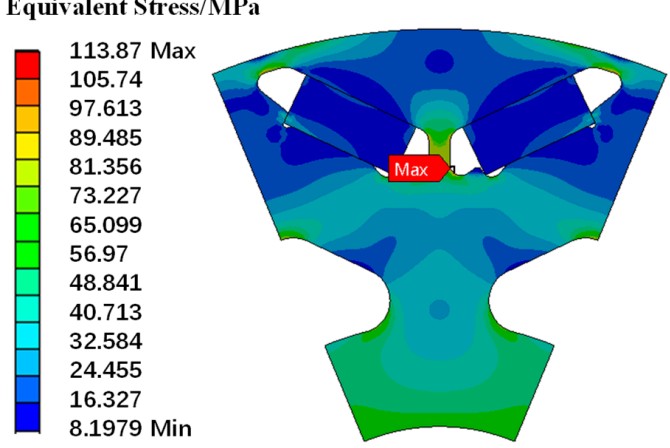

**Figure 13.** Rotor stress distribution under a structural field.

The above picture shows the FEA results when the rotational speed reaches a peak value of 8907 rpm at 501 s of the working condition. It can be found that the concentrated

stress of the rotor is at the center magnetic bridge of the rotor, the maximum stress value is 113.87 MPa, and the maximum radial displacement is 0.0145 mm.

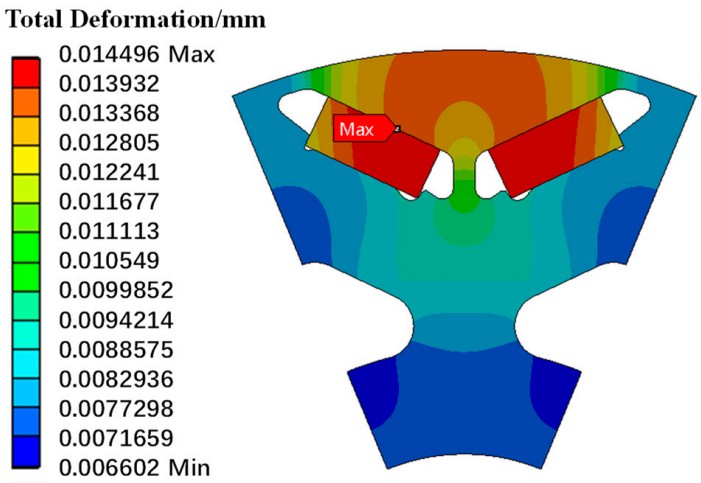

**Figure 14.** Rotor deformation under a structural field.

### 3.5. Load Analysis under Coupled Thermomechanical Fields

During motor operation, internal heat is generated, resulting in a rise in rotor temperature. Due to the different thermal expansion coefficients of materials and the constrained extrusion of the structure, thermal stress is generated inside the rotor. Therefore, the rotor temperature distribution obtained from the thermal field simulation was imported into the structural field, and the rotor model was loaded with the speed under test conditions to obtain the FEA results of the rotor under coupled thermomechanical stress. Figures 15 and 16 also show the simulation results for a rate of 8907 rpm.

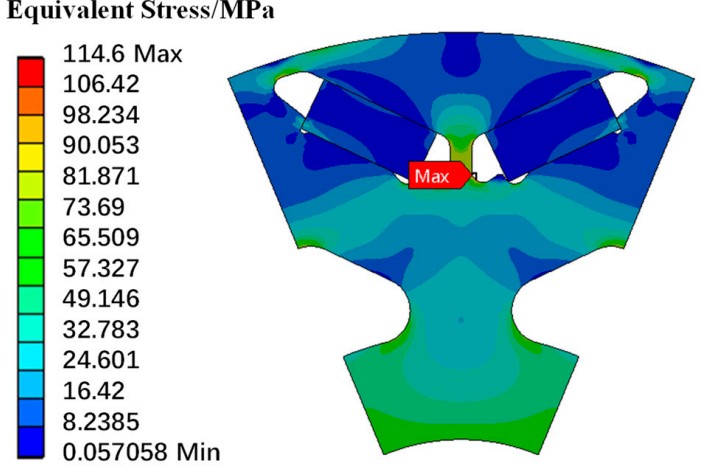

**Figure 15.** Rotor stress distribution under coupled thermomechanical fields.

The above picture shows the FEA results when the rotational speed reaches a peak value of 8907 rpm at 501 s of the working condition. It can be found that the concentrated stress of the rotor is at the center magnetic bridge of the rotor, the maximum stress value is 113.87 MPa, and the maximum radial displacement is 0.0145 mm.

The FEA analysis results show that the maximum stress of the rotor is also located at the central magnetic bridge under the action of thermomechanical stress coupling, with a maximum stress of 114.6 MPa, which is less than a 1 MPa increase compared to the centrifugal force alone. The maximum rotor deformation is 0.049 mm, which is 3.4 times greater than that in a single mechanical field. The air gap between the rotor and the stator is 0.8 mm, which is within a reasonable range.

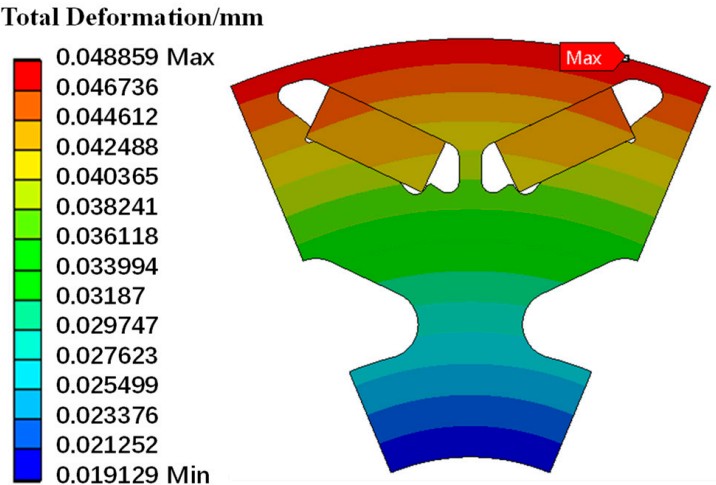

**Total Deformation/mm**

**Figure 16.** Rotor deformation under coupled thermomechanical fields.

Figure 17 shows the comparative stress variation curves at the central magnetic bridge for the structural and thermomechanical fields. It can be seen from the figure that the mechanical stress curve and the stress curve under the thermal-mechanical coupling are almost coincident, and only under the working condition of a lower speed will the thermal-mechanical stress be immensely more significant than the mechanical stress. At low-speed operating conditions, the overall temperature of the motor will decrease, while the thermal stress at the magnetic bridge will increase due to the difference in thermal conductivity and heat generation rate between the rotor and the permanent magnet. The mechanical stress is proportional to the square of the speed and has little effect on the magnetic bridge.

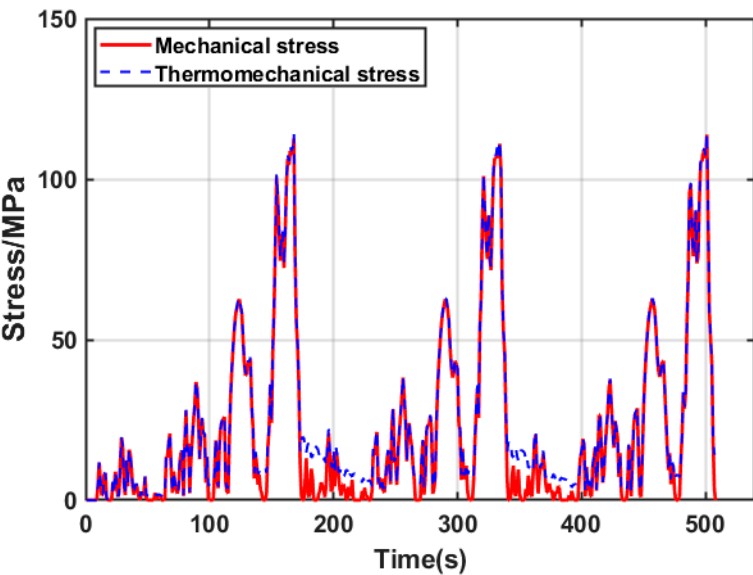

**Figure 17.** Stress comparison curve at the magnetic bridge.

*3.6. Failure Mode Analysis and Validation*

The contribution of simulated loads to magnetic bridge damage was analyzed for multiple physical fields, specifically at the rated extreme operating speed of 8907 rpm. The load simulation analysis under the electromagnetic, structural, and thermomechanical coupling fields indicates that the electromagnetic stress at the magnetic bridge is 1.5 MPa, the mechanical stress under the structural field is 113.87 MPa, and the combined stress under the thermomechanical coupling is 114.6 MPa. The electromagnetic and thermal stresses at the central magnetic bridge of a V-shaped permanent magnet synchronous motor are low.

Furthermore, it was determined that damage to the magnetic bridge under electromagnetic, thermal, mechanical, and thermomechanical stress accounts for a proportion of the damage under electrical/thermal/mechanical stress. As shown in Figure 18, the contribution ratio of electromagnetic and thermal stress to damage is less than 0.1; the contribution of single mechanical stress to damage reaches 0.95. Therefore, the main failure load of the magnetic bridge is mechanical stress, and its failure mode is mainly fatigue failure.

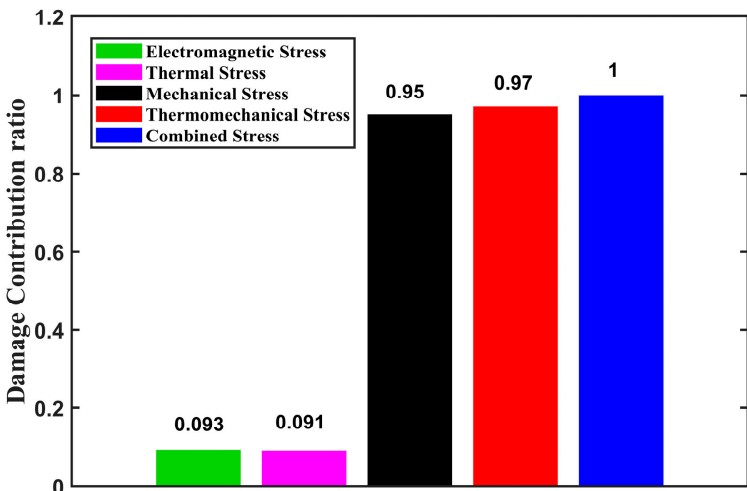

**Figure 18.** Damage contribution ratio under different stress loads.

Figure 19 depicts the failure diagram of the rotor laminations of the motor. It is evident that the rotor laminations in the middle area exhibit a single-piece protrusion phenomenon. The centrifugal force at high speeds causes damage to the central magnetic bridge of the rotor before the rotor can protrude as a single piece, which is consistent with the simulation results.

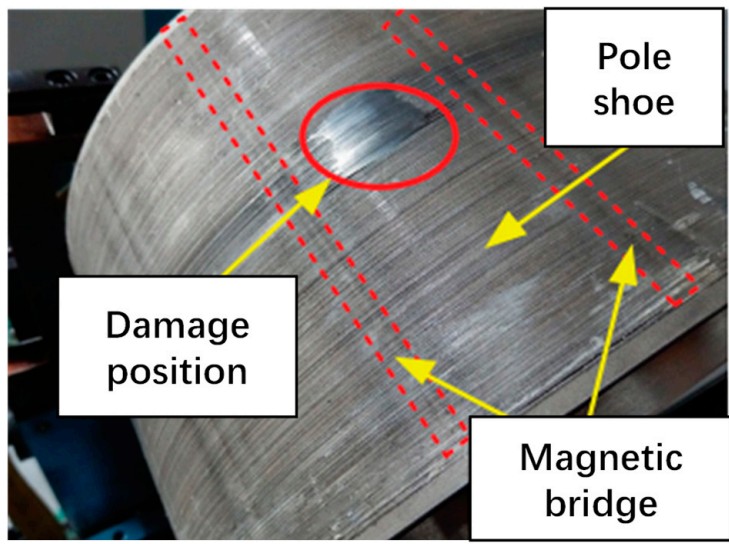

**Figure 19.** The failure diagram of the rotor.

## 4. Fatigue Analysis and Life Modeling

### 4.1. Fatigue Testing

The rotor studied in this paper is a multi-layer punch-laminated PMSM rotor. The central magnetic bridge of the rotor is a location prone to stress concentration and represents a weak point for rotor strength and fatigue design. To perform a fatigue analysis of the motor rotor, it is necessary to obtain an S–N curve for the rotor material. The material

used for the rotor laminations is a non-oriented electrical steel plate produced by Baosteel, with a thickness of 0.3 mm, a silicon content of approximately 3.1%, and a width of 3 mm for its central magnetic spacer bridge. In this study, fatigue specimens of the same size were selected for testing to approximate the actual component and reduce the effect of the structure on its life, as shown in Figure 20.

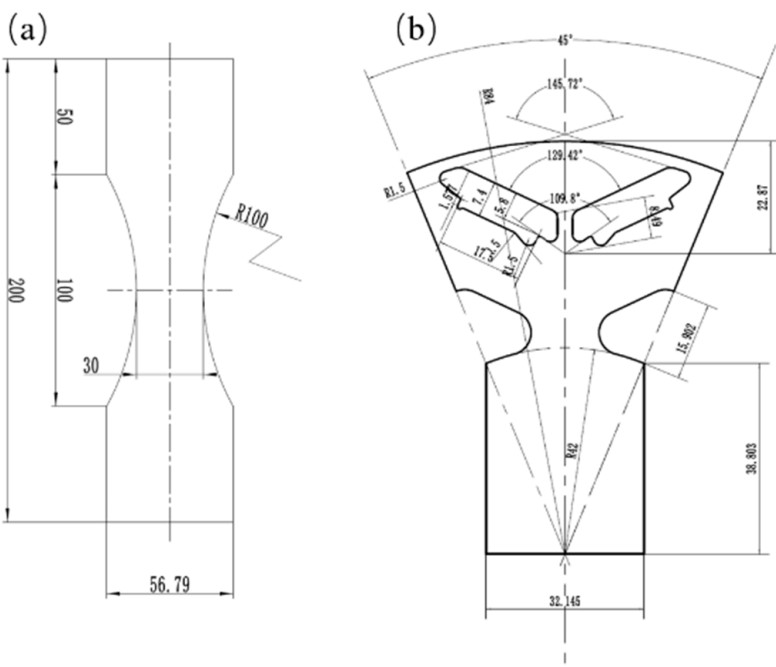

**Figure 20.** B30AVH1500 standard fatigue specimen (**a**) and rotor fatigue specimens (**b**) (unit: mm).

All fatigue tests were performed on a 100 kN RUMUL electromagnetic resonance fatigue tester [21], as illustrated in Figure 21. The stress ratio R was set to 0.1, the test frequency was approximately 40 Hz, and tests were conducted using secondary stress levels with maximum stresses of 450 MPa and 350 MPa. Fatigue tests were conducted at both 20 °C and 150 °C, as the S–N curve of the rotor laminations also varies at different temperatures, allowing for further consideration of the effect of temperature on rotor life.

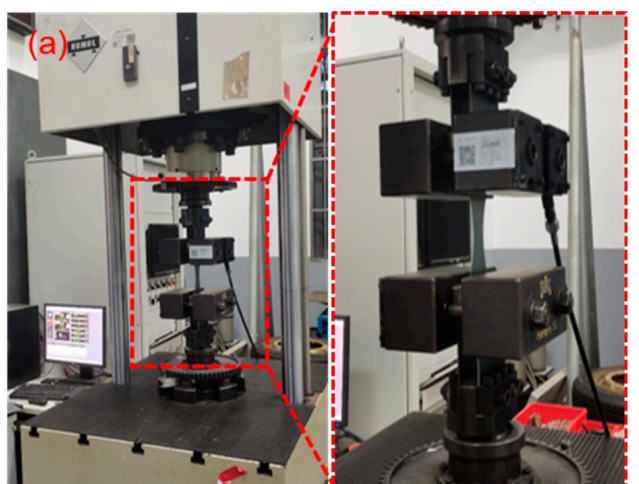

(**a**) Standard fatigue specimen

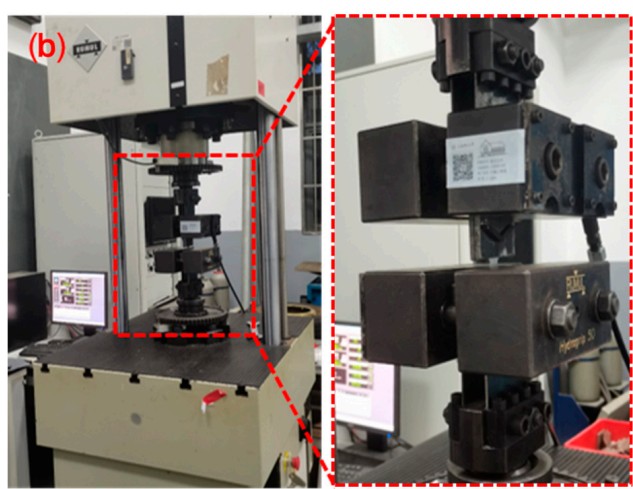

(**b**) Rotor fatigue specimens

**Figure 21.** Fatigue specimen test diagram.

The fatigue test results of the standard specimen are shown in Figure 22:

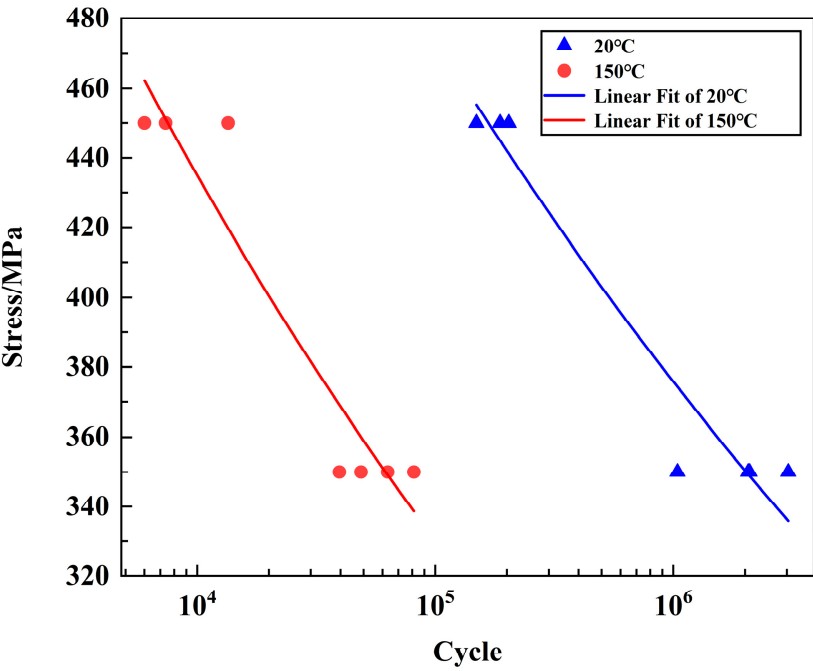

**Figure 22.** Standard fatigue specimen life curve.

The S–N curve obtained by fitting at 22 °C is as follows:

$$S_a = 1037.55307 - 110.87374 \lg N_f \tag{7}$$

The S–N curve obtained by fitting at 150 °C is as follows:

$$S_a = 900.2793 - 118.5052 \lg N_f \tag{8}$$

The S–N curve is fitted using a logarithmic coordinate system, where the horizontal axis represents stress levels and the vertical axis represents the logarithmic value of the number of cycles. Under this form, the S–N curve typically presents a linear shape, where the slope k and intercept C of the straight line are both material parameters. By extracting the values of the slope k and intercept C of the material's S–N curve under ambient and high-temperature conditions, the S–N curve under other temperature conditions can be extrapolated using linear interpolation. The resulting graph is shown in Figure 23 below:

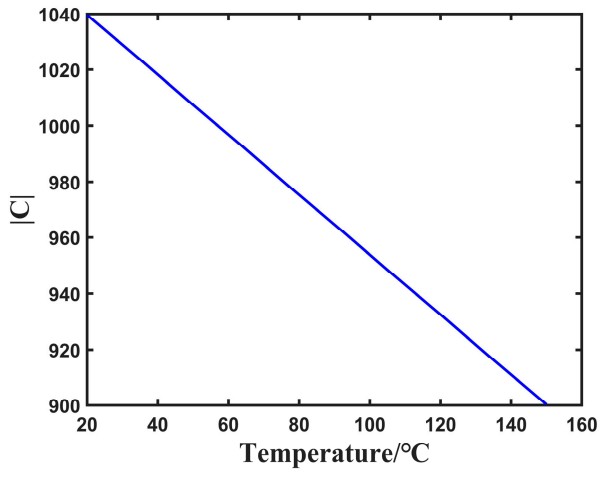

(**a**) Intercept- Temperature curve

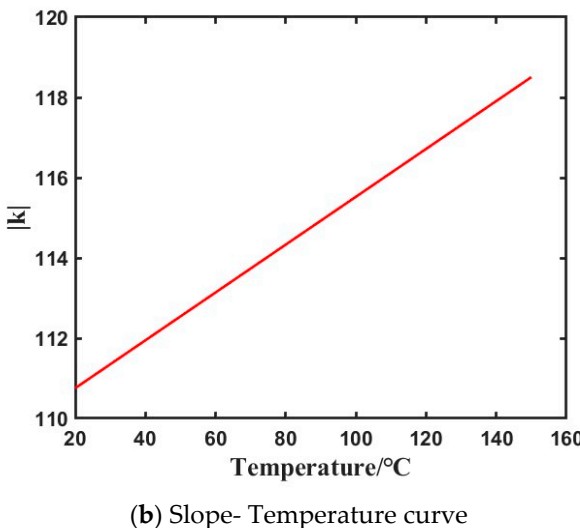

(**b**) Slope- Temperature curve

**Figure 23.** Distribution of S–N slope and intercept at different temperatures.

When conducting fatigue tests on rotor blades, the fatigue specimens need to simulate the centrifugal force load of the rotor under high-speed rotation. As shown in Figure 24, the centrifugal force is simulated by applying tensile loads on the test machine. Three typical operating conditions are selected for loading during the test process: 400 N, 500 N, and 600 N. Figure 24 shows the stress distribution of the motor rotor laminations under a tensile load of 600 N, and the presence of the clamping plate above the rotor facilitates the addition of tensile loads in the finite element setting. The analysis results show that the maximum stress occurs at the center inter-pole bridge, with a magnitude of 462 MPa. The stress levels corresponding to the weak positions under 500 N and 400 N are 425 MPa and 388 MPa, respectively.

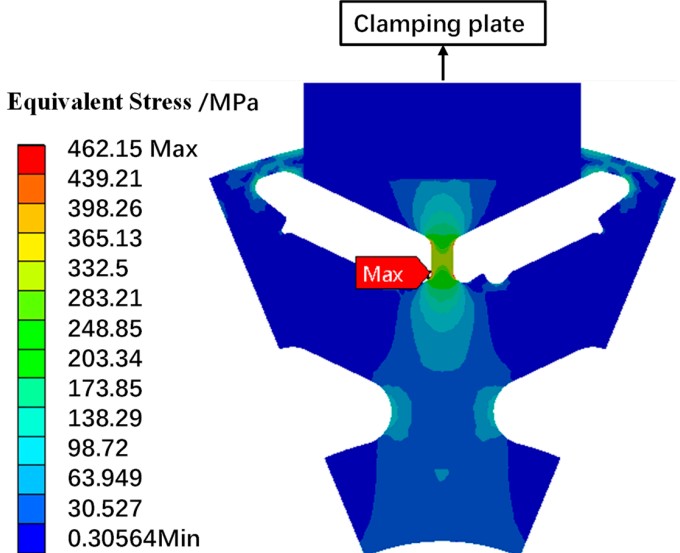

**Figure 24.** Rotor stress distribution under a tensile load of 600 N.

The experimental results of the fatigue specimens with rotor structural features are presented in Figure 25, with lifetimes of approximately $10^5$, $2.2 \times 10^5$ and $6 \times 10^5$ cycles under the loading conditions of 600 N, 500 N, and 400 N, respectively. The lifespan results are essentially similar when compared to the standard fatigue specimens, as detailed in Table 3.

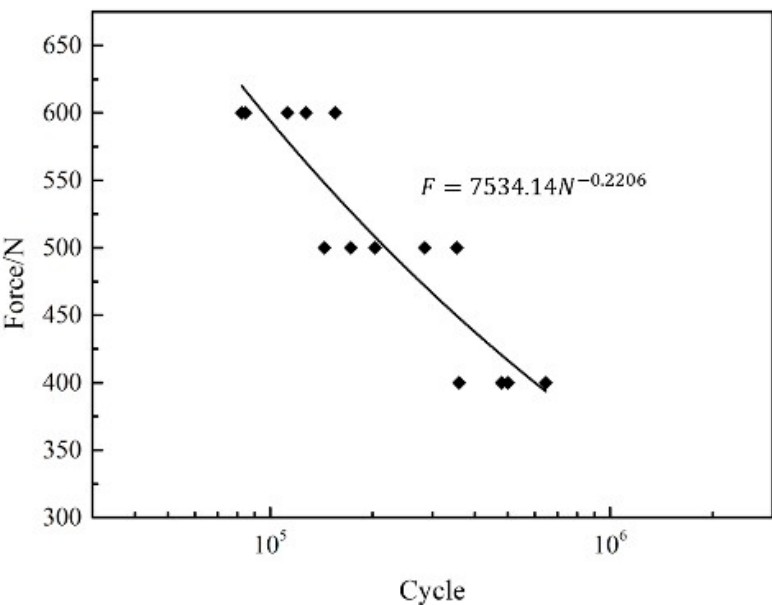

**Figure 25.** Fatigue life curve of rotor structure test specimens.

**Table 3.** Comparison of life between rotor fatigue specimens and standard fatigue specimens.

| Load and Life | Value | Value | Value |
|---|---|---|---|
| Force/N | 400 | 500 | 600 |
| Maximum stress/MPa | 388 | 425 | 462 |
| Rotor fatigue specimen life | $6.0 \times 10^5$ | $2.2 \times 10^5$ | $1.0 \times 10^5$ |
| Standard fatigue specimen life | $7.2 \times 10^5$ | $3.3 \times 10^5$ | $1.5 \times 10^5$ |

*4.2. Fatigue Aging Damage Model*

The internal rotor's temperature constantly changes during the operation of the motor, and when the temperature rises, the fatigue strength of the rotor decreases. The temperature load histories obtained from the simulation of the magnetic bridge out in the thermal field are distributed jointly with the stress load histories obtained under structural field coupling, and the time duration at each temperature level and stress level is counted. According to the S–N curve of the material at different temperatures, the damage analysis of the rotor is carried out.

The number of cycles $n_{ij}$ with temperature $T$ at level i and stress $\sigma_j$ at level j is obtained after counting through the joint distribution. The S–N curve for the rotor at temperature $T_i$ can be seen in Figure 23 and the fatigue life $N_{ij}$ at this temperature and stress level can be calculated. According to the Miner linear cumulative damage criterion [22], the damage $D_s$ incurred by the motor rotor is calculated as follows:

$$N_{ij} = 10^{\frac{|C|_{T_i} - \sigma_j}{|k|_{T_i}}} \tag{9}$$

$$D_s = \sum_{i=1}^{n} \sum_{j=1}^{n} \frac{n_{ij}}{N_{ij}} \tag{10}$$

where $|C|_{T_i}$ is the rotor material intercept at a temperature of $T_i$ and $|k|_{T_i}$ is the rotor material slope at a temperature of $T_i$.

*4.3. Fatigue Damage Model under Alternating Loads*

The frequent acceleration and deceleration of a vehicle during driving and the high-frequency alternation of rotational speeds can have a damaging effect on the motor rotor that cannot be ignored. The number of cycles at each level of load range and mean value is extracted by rainflow counting of the speed–time course using the range–mean counting method [23], which in turn allows us to perform a damage analysis of the load counting results at different operating conditions.

The number of cycles $n_{R,i}$ at a torque range $S_{R,i}$ of level *i* is obtained by counting the rainflow cycles of the torque. Combined with the material S–N load life curve equation $S_R^m \cdot N_f = C$, the fatigue life $N$ at level *i* of the torque variation can be derived. According to the Miner linear cumulative damage criterion, the damage $D_s$ generated by the motor rotor can be obtained, and the calculation formula is as follows:

$$D_s = \sum_{i=1}^{N_k} \frac{n_{R,i}}{N_{f,i}} \tag{11}$$

**5. Compilation of Accelerated Test Load Spectrum**

*5.1. The Whole Life Cycle Damage Targets*

The durability assessment of an electric vehicle motor rotor is typically carried out in bench tests using an accelerated test load spectrum to conduct a rapid durability test. Reasonable test load spectra are essential in product performance assessments, test verification, and product design. Typical operating conditions and life-cycle durability objectives complement each other in the reliability evaluation of products. The objectives of the

reliability evaluation must ultimately be broken down into individual operating conditions, the cumulative effect of which forms the overall objective. To determine the life-cycle damage target for the rotor, the total damage to the rotor under the 300,000 km working load needs to be specified, as shown in Table 4.

**Table 4.** Damage targets under different failure modes.

| Failure Mode | Damage Targets |
|---|---|
| Fatigue aging coupling | 0.24 |
| Alternating loads fatigue | 0.013 |

### 5.2. Selection of Reliability Test Conditions

The experimental conditions for the rotor fatigue aging coupled failure mode can be selected based on the overall distribution of temperature and speed. By jointly counting the temperature and speed distribution of all user data, the frequency under different load levels is calculated and shown in Figure 26a. Using Equations (9) and (10), the damage distribution under different load levels is calculated and shown in Figure 26b. Three reliable test conditions are selected: low speed/low temperature, medium speed/medium temperature, and high speed/high temperature, each lasting 100 s. The specific test condition points are shown in Table 5. It can be seen from the frequency distribution and damage distribution curves that the proportion of low-speed working conditions and the corresponding damage ratio are relatively high. At this working condition, the rotor temperature is mainly distributed between 45 °C and 69 °C; therefore, the first working condition point selected is 640 rpm and 60 °C. When the speed is in the range of 4072 rpm to 8357 rpm, the motor is in the medium-speed working condition. The overall damage distribution of the rotor under this condition is not only high but also relatively uniform. At this point, the rotor temperature is mainly distributed between 69 °C and 112 °C; therefore, the second working condition point selected is 6640 rpm and 90 °C. The core of preparing the acceleration test load spectrum is to transfer the damage from small loads to large loads based on the principle of damage equivalence in order to shorten the test time. Therefore, although the frequency and damage ratio of the high-speed working condition are not high, the third working condition point selected is 10,070 rpm and 110 °C, which belongs to a relatively harsh working condition.

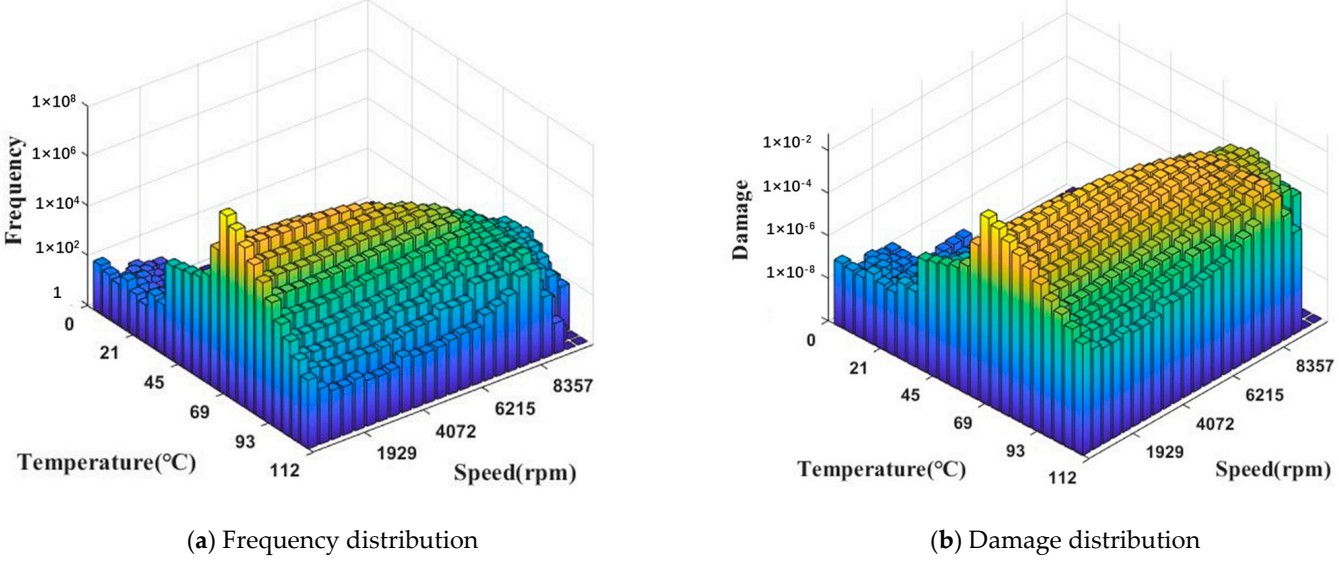

(**a**) Frequency distribution　　　　　　　　　　　　　　(**b**) Damage distribution

**Figure 26.** Speed–temperature frequency and damage distribution.

**Table 5.** Typical working condition points for fatigue aging and coupling failure modes.

| Condition | Speed (rpm) | Temperature (°C) | Cycle Time for a Single Condition |
| --- | --- | --- | --- |
| 1 | 640 | 60 | 100 s |
| 2 | 10,070 | 110 | 100 s |
| 3 | 6640 | 90 | 100 s |

The typical operating conditions for the rotor under alternating cyclic fatigue failure mode can be selected based on the overall distribution of the mean and range of the rotor speed. By performing rainflow counting on the rotor speed data for all users, the frequency distribution under different means and ranges of rotor speeds is obtained, as shown in Figure 27a. The damage distribution under different means and ranges of rotor speeds can be calculated using Formula (11), as shown in Figure 27b. Based on the distribution of the load, the range of the rotational speed was roughly divided into three zones: (0, 3500) for the low-rotational-speed range, (3500, 7000) for the medium-rotational-speed range, and (7000, 10,000) for the high-rotational-speed range. One larger and more frequently occurring operating condition was selected from each range, and the final operating condition points were selected as shown in Table 6.

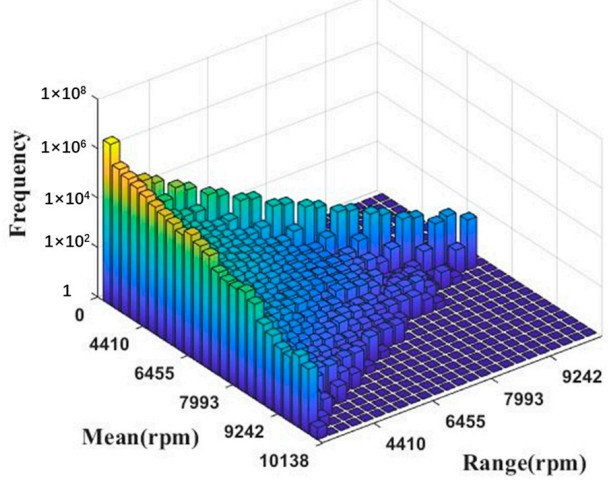

(**a**) Frequency distribution

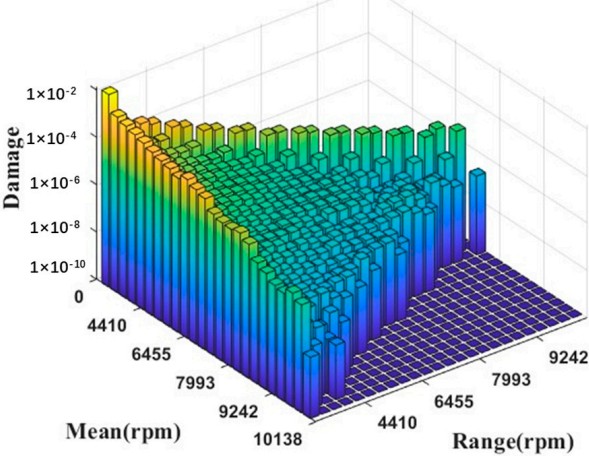

(**b**) Damage distribution

**Figure 27.** Rotation range—mean frequency and damage distribution.

**Table 6.** Typical operating conditions for variable amplitude fatigue failure mode of motor rotor.

| Condition | Range (rpm) | Temperature (°C) | Cycle Time for a Single Condition |
| --- | --- | --- | --- |
| 4 | 3320 | 60 | 100 s |
| 5 | 9500 | 60 | 100 s |
| 6 | 6470 | 60 | 100 s |

As shown in Figure 26, 60 °C is the most frequent temperature point, and this temperature point is also a relatively balanced result. Therefore, setting the temperature of all three operating conditions to 60 °C is reasonable. Each operating condition lasts for 100 s, and the rotational speed rises from 0 to the maximum range and then drops to 0 for one cycle, which lasts for 4 s. One complete operating condition lasts for 25 cycles, as shown in Figure 28.

### 5.3. Optimization of the Number of Cycles

The accelerated test load spectrum of the motor rotor is compiled to reproduce the effects of damage under different road sections simultaneously when reproducing the

300,000 km total damage. It is crucial to reproduce damage at different constant speed temperatures as well as damage at different speed range variations. Therefore, the determination of the cycle times of the whole test condition is a typical multivariate and multi-objective optimization problem [24].

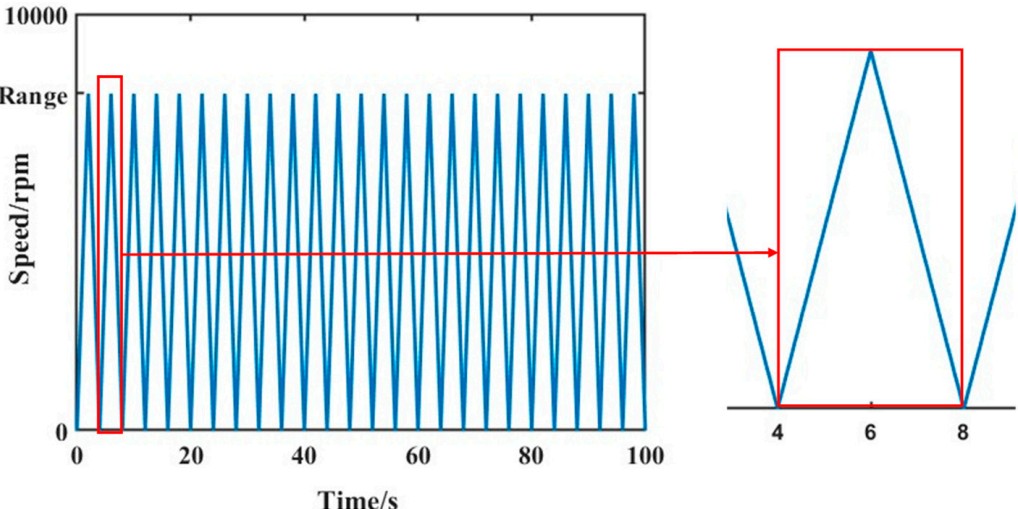

**Figure 28.** Speed-changing operating condition.

NSGA-II is one of the most commonly used multi-objective genetic algorithms in engineering applications. It is an optimization algorithm based on the first-generation algorithm and has the advantages of fast convergence, wide applicability, and good robustness. The specific solution process of the NSGA-II algorithm is shown in Figure 29.

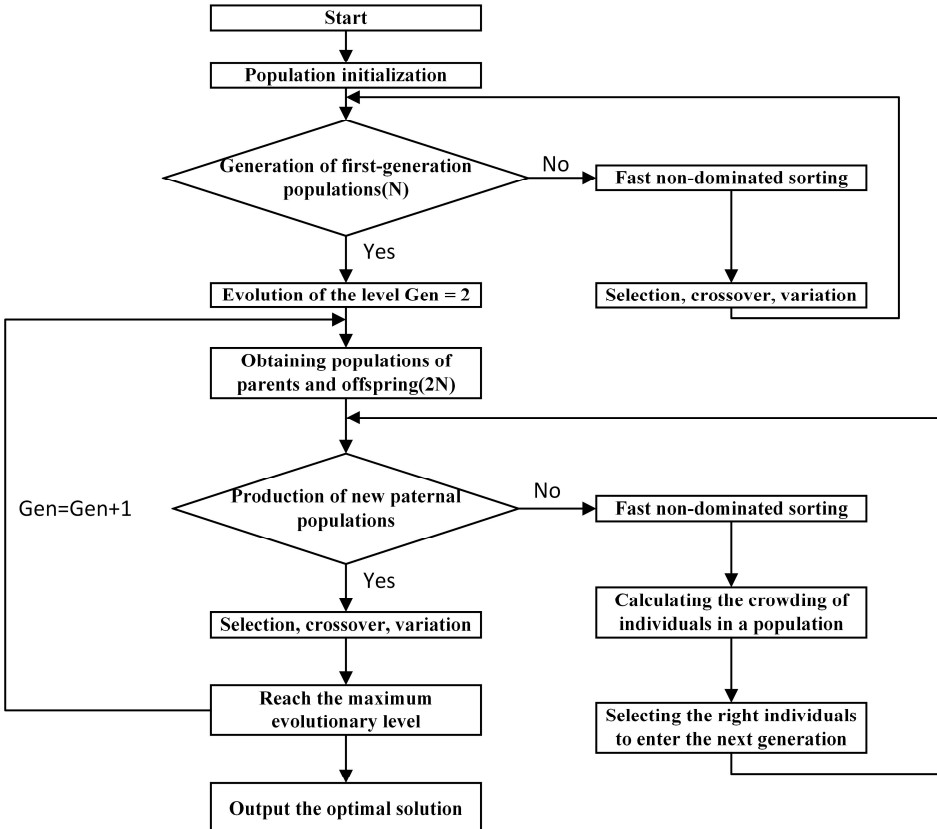

**Figure 29.** NSGA-II algorithm principle flowchart.

The fatigue damage caused to the rotor by the six typical bench test conditions selected in this paper form a matrix of fatigue damage factors, defined as $D_{n1}, D_{n2}, \cdots, D_{n6}$. Damage to the rotor at different constant speeds and temperature and speed ranges are $D_1, D_2$. A multi-objective optimization algorithm was used to calculate the number of cycles for each reliability test segment covering the multi-damage objective, calculated as follows:

$$\begin{bmatrix} D_{11} & D_{12} & D_{13} & D_{14} & D_{15} & D_{16} \\ D_{21} & D_{22} & D_{23} & D_{24} & D_{25} & D_{26} \end{bmatrix} \begin{bmatrix} X_1 \\ X_2 \\ \vdots \\ X_6 \end{bmatrix} = \begin{bmatrix} D_1 \\ D_2 \end{bmatrix} \tag{12}$$

where $D_1, D_2$ are the six typical operating conditions that cause damage to the motor rotor; and $X_1, X_2, \cdots, X_6$ are the number of cycles of the six reliability test conditions.

The total time for a single test cycle was 4600 s. The number of cycles for the two damage targets of the motor rotor for the six test conditions was obtained by a multi-objective optimization solution, as shown in the following table.

The number of cycles for the test conditions in Tables 7 and 8 was allocated according to the proportion of the total time spent in a single operating condition, and the final load spectrum for the accelerated test of the motor rotor was obtained, as shown in Figure 30. The total time for a single cycle of the load spectrum was 4600 s, and the total test time to reproduce the rotor life cycle damage target was 716 h. As shown in Figure 31, the damage recovery ratios for both failure modes were close to one, indicating that the accelerated test load spectrum can reproduce the total damage over 300,000 km. Ultimately, the accelerated test load spectrum operating for 716 h can reproduce the user's operating time to 11,571 h at 300,000 km, with an acceleration factor of 16.16.

**Table 7.** Number of cycles for each test condition in fatigue aging failure mode.

| Condition | Speed (rpm) | Temperature (°C) | Total Number of Cycles | Number of Cycles of Test Conditions in One Cycle |
|-----------|-------------|------------------|------------------------|---------------------------------------------------|
| 1 | 640 | 60 | 2241 | 4 |
| 2 | 10,070 | 110 | 13,439 | 24 |
| 3 | 6640 | 90 | 4480 | 8 |

**Table 8.** Number of cycles for each test condition in alternating loads fatigue mode.

| Condition | Range (rpm) | Temperature (°C) | Total Number of Cycles | Number of Cycles of Test Conditions in One Cycle |
|-----------|-------------|------------------|------------------------|---------------------------------------------------|
| 4 | 3320 | 60 | 562 | 1 |
| 5 | 9500 | 60 | 2238 | 4 |
| 6 | 6470 | 60 | 2800 | 5 |

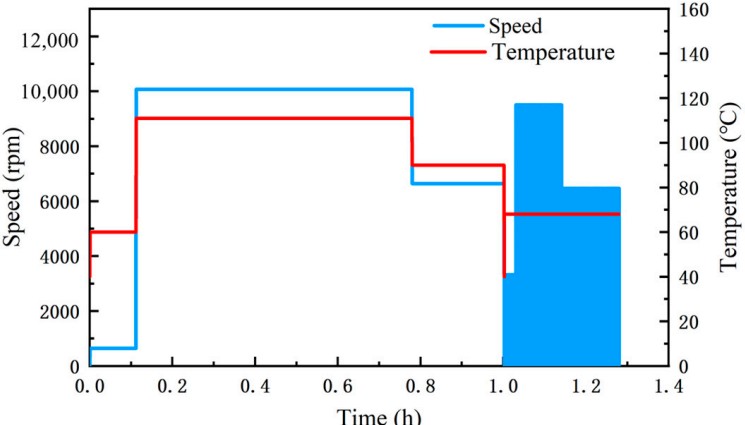

**Figure 30.** Load spectrum for accelerated tests.

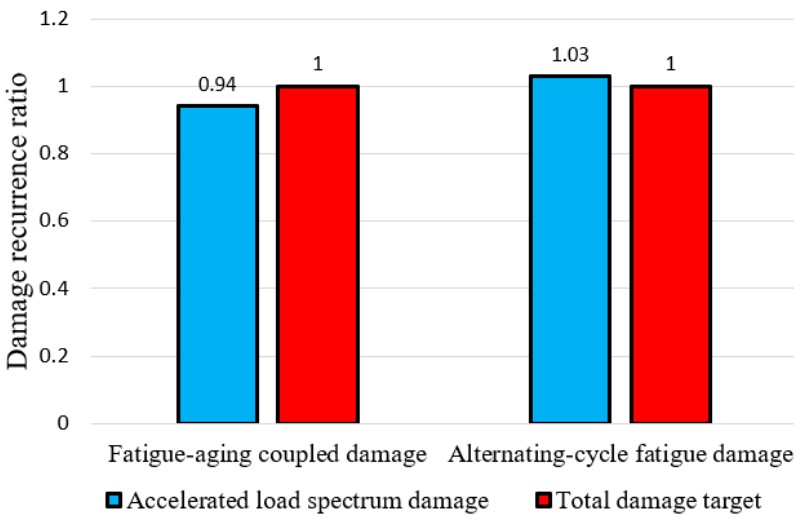

**Figure 31.** Damage recovery ratios for both failure modes.

## 6. Conclusions

In this paper, we analyze the failure mode of the rotor under multi-physical loads and construct an accelerated reliability test load spectrum for the magnetic bridge, which is prone to monolithic protrusion and bridge fracture during the actual operation of electric vehicle drive motors. Our main conclusions are as follows.

(1) We propose a method for analyzing the failure mode of the magnetic bridge and constructing an accelerated reliability test cycle under multi-physics fields. This method includes establishing a physical model of the rotor failure, simulating and verifying multi-physics field loads, analyzing magnetic bridge damage, establishing life models, determining whole life cycle damage targets, selecting reliability test conditions, optimizing the number of cycles, and compiling an accelerated test load spectrum.

(2) In the multi-physics field load simulations, we identify the stress hazard points in the rotor at the magnetic bridges. The maximum point of electromagnetic stress on the rotor surface occurs at the intersection of the permanent magnet and the rotor slot. Under severe operating conditions, the stress magnitude at the spacer bridge is less than 2 MPa, rendering the effect of electromagnetic forces negligible. Additionally, the highest rotor temperatures occur near the magnets in the simulation analysis by thermal field loading. The rotor temperature decreases from the inside to the outside along the permanent magnet region, and the temperature rise of the stator has less influence on the rotor due to the air gap.

(3) We define the whole life cycle damage targets of the rotor magnetic bridge under fatigue failure and fatigue aging coupled failure. We construct the accelerated test cycle conditions, which include steady-state conditions and alternating conditions. We use a multi-objective optimization algorithm to determine the optimal number of cycles of working conditions and to construct a load spectrum for accelerated tests of magnetic bridges considering multiple failure modes. Ultimately, we develop an accelerated test load spectrum operating for 716 h, which can reproduce the user's operating time of 11,571 h at 300,000 km, and the acceleration factor is 16.16.

**Author Contributions:** Conceptualization, S.W., J.S., L.Z. and L.L.; methodology, S.W., J.S., L.Z., L.L., Z.W. and X.W.; validation, S.W., J.S. and D.W.; formal analysis, S.W. and L.Z.; investigation, J.S., L.Z. and Z.W.; resources, D.W and X.W.; data curation, S.W. and L.L.; writing—original draft preparation, S.W.; writing—review & editing, S.W., J.S. and L.L.; visualization, S.W.; supervision, J.S.; project administration, S.W. and L.Z.; funding acquisition, S.W. All authors have read and agreed to the published version of the manuscript.

**Funding:** This research received no external funding.

**Institutional Review Board Statement:** Not applicable.

**Informed Consent Statement:** Not applicable.

**Data Availability Statement:** Not applicable.

**Conflicts of Interest:** The authors declare no conflict of interest.

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
