# Peer review of "Failure Analysis and Accelerated Test Development for Rotor Magnetic Bridge of Electric Vehicle Drive Motor"

_applsci, doi:10.3390/app13084753_

Round 1

Reviewer 1 Report

Congratulations for your work. I think that this is a good work grouping different aspects of mechanical design and clarifying the mechanical aspects of rotor design.

Author Response

Dear Reviewer,

Thank you very much for taking the time to review my manuscript titled "Failure analysis and accelerated test development for rotor magnetic bridge of electric vehicle drive motor" submitted to Applied Sciences. I sincerely appreciate your positive feedback and kind words regarding my work.

I am pleased to hear that you found the paper to be a good contribution to the field, particularly in terms of the mechanical design and rotor design aspects. Your feedback will undoubtedly help me to improve the manuscript further.

Thank you again for your time and effort in reviewing my manuscript.

Best regards,Le Li

Reviewer 2 Report

This article presents the failure analysis and accelerated test development for rotor magnetic bridge of electric vehicle drive motor. The idea presented in the paper is good. However, the article needs minor revision. Some comments and suggestions are given here:

- Organize the article according to the MDPI journal template.

- In introduction, the main contribution and originality should be explained in more detail.

- Improve the quality of the results presented in Figure 18.

- The interpretation of some results needs to be improved (Figures 17 and 23).

Author Response

Dear Reviewer,

Thank you for taking the time to review our manuscript titled "Failure analysis and accelerated test development for rotor magnetic bridge of electric vehicle drive motor" submitted to Applied Sciences. We appreciate your valuable feedback and suggestions, which have helped us improve the quality of our work.

In response to your first comment, we have completely revised the manuscript's layout, including the abstract, text, and figures, to comply with the MDPI journal template. We will communicate with the editor to ensure that the final version is consistent with the journal's guidelines.

Regarding your second comment, we have added a more detailed explanation of the main contributions and originality of our study in the Introduction section. The added content is highlighted in blue font.

Regarding your third comment, we have redrawn Figure 18, "Damage contribution ratio under different stress loads," to improve the presentation quality of the results.

Regarding your fourth comment, we have added detailed explanations to Figure 17, "Stress comparison curve at the magnetic bridge," and Figures 26 and 27 (formerly Figures 23 and 24), "Speed-temperature frequency and damage distribution" and "Rotation range - mean frequency and damage distribution," respectively. The added content is highlighted in blue font. We hope that these modifications have addressed your concerns and improved the clarity of our findings.

Thank you again for your constructive feedback and helpful suggestions, which have contributed to the enhancement of our manuscript. We look forward to your continued support.

Best regards, Le Li

Reviewer 3 Report

1. The numerical value on the left side of the surface force strain diagram in Figure 4 is too large to clearly represent the distribution of surface force.

2. For Figure 19, the previous article mentioned that considering the uniform distribution of the rotor along the axial direction, the simulation uses a two-dimensional planar model. However, in this figure, we can clearly see that the failure of the rotor only occurs in the middle of the axial direction, not in the entire axial direction. The author should explain this.

3. "Based on the slope k and intercept C values for these two operating conditions, the S-N curves for different temperature operating conditions were extrapolated using the linear difference method, as shown in Figure. 22." was mentioned earlier, but we can see that the curve in Figure 22 is not an S-N curve, and the author should explain this.

4. During fatigue testing, secondary stress levels with maximum stresses of 450Mpa and 350Mpa were used for testing, as well as fatigue testing at 20 ℃ and 150 ℃. What is the basis for selecting the experimental conditions?

5. In chapter 4.1, when conducting fatigue tests, fatigue samples were selected for testing, which could not assess the gap between this and the actual components. Therefore, the referential nature of the experimental results was reduced.

Author Response

Reply:

Dear Reviewer,

Thank you for your helpful comments on our manuscript titled "Failure analysis and accelerated test development for rotor magnetic bridge of electric vehicle drive motor" submitted to Applied Sciences. We have made the following modifications to address your concerns:

  1. In response to the comment about the numerical value on the left side of the surface force strain diagram in Figure 4, we would like to explain that the short and dense blue arrow represents the surface electromagnetic force density of the rotor, which has a value of 48.5 N/m^2, which is a normal value. The red arrow represents the electromagnetic force density at the winding coil, which is typically in the order of kilonewtons and acts on the coil's ampere force. As the wire's diameter is about 0.8 mm, the magnitude of the result when the kilonewtons ampere force is divided by the wire's cross-sectional area is approximately 1E+10. As the study focuses on the rotor of the electric motor, we mainly analyze the specific stress distribution of the rotor's different parts by observing the overall distribution of the blue arrows.
  2. With regard to your second comment, we analyzed the failure of the motor rotor mainly from the perspective of stress under multiple physical fields. There are also other uncontrollable factors such as mechanical vibration during the actual operation of the motor rotor. In general, the fracture of the rotor magnetic bridge may also occur on both sides. Stress under multiple physical fields is the main cause of magnetic bridge fracture and protrusion.
  3. Regarding your third comment, we used a logarithmic coordinate system to fit the material S-N curve, which usually appears as a straight line. The slope k and intercept C in the S-N curve are key parameters that determine the material performance. Therefore, we extracted the slope k and intercept C values of the material S-N curve under two temperature operating conditions mentioned earlier, and extrapolated the S-N curve under other temperature operating conditions using the linear difference method. Since equations (9) and (10) require the slope k and intercept C values of the material under different temperatures for damage calculation, Figure 22 (now modified to Figure 23) shows the variation of the slope and intercept of the material's S-N curve at different temperatures, rather than the S-N curve at a specific temperature. We have also modified the explanation of Figure 23 accordingly.
  4. Regarding the basis for selecting the experimental conditions during fatigue testing, we selected the maximum stress levels of 450 MPa and 350 MPa based on 70% and 90% of the material's yield strength, respectively. We chose 20°C as it is an easily operable temperature, while 150°C was chosen because it represents the highest temperature inside the motor.
  5. In chapter 4.1, we agree that testing fatigue samples may not fully assess the gap between them and the actual components. To address this concern, we conducted fatigue tests on actual structural samples as well. Due to time and resource constraints, only tests at room temperature were performed on the rotor structural samples. However, combining finite element simulation and experimental results, we found that the fatigue life results of the rotor structural samples were similar to those of the standard fatigue samples. We have added this information to the manuscript, and the added content is highlighted in blue font.

Thank you again for your constructive feedback and helpful suggestions, which have contributed to the enhancement of our manuscript. We look forward to your continued support.

Best regards, Le Li

Reviewer 4 Report

This paper presents a failure analysis and accelerated test development for rotor magnetic bridge of electric vehicle drive motor. I believe the paper is well written and the topic is if interest. Moreover, several simulation as well as experimental tests are provided. Hence, I congratulate the authors on their efforts. I believe, however, that some aspects of the manuscript can be improved before it can be considered for publication.

Please find my comments as follows:

1.      English mistakes can be found in the manuscript, for example, see Figure 19. I suggest the authors to go over the paper once more and address them.

2.      The contributions of the paper are not well presented in the end of the introduction. I believe they should be explained with more detail. Consider enumerating them or using bullet points as it has been done in the cover letter.

3.      Looking at Figure 3, the speed profile seems like a periodic profile but has a somewhat unusual characteristic. Is this data from a certain application? If possible, include this information.

4.      Some aspects regarding the multiobjective optimization are not clear. What is the algorithm used? What are the cost functions? Consider expanding the explanation or even including a small flowchart of the procedure.

5.      Figure 25 is not clear to me. What are the blue blocks at the end of the figure? Is that the speed line changing very rapidly?

6.      Lastly, when possible, consider using vector graphics (eps, pdf), as some of the figures get pixelated when reasonable zoom is applied.

Author Response

Thank you for your constructive feedback on our manuscript titled "Failure analysis and accelerated test development for rotor magnetic bridge of electric vehicle drive motor". We have carefully considered your suggestions and made the necessary revisions to address your concerns.

  1. We have carefully reviewed the manuscript and corrected all the English mistakes, including the one in Figure 19.
  2. We have added a detailed explanation of the contributions of the paper at the end of the introduction, using bullet points to make them more clear and concise.
  3. We would like to clarify that the speed profile data presented in Figure 3 is obtained from a specific application, which is a test bench running condition provided by our partner company.
  4. We have expanded our explanation of the multiobjective optimization algorithm used and included a small flowchart of the procedure in Figure 29.
  5. We would like to clarify that the blue blocks at the end of the figure represent the process of alternating speed, with a period of 4 seconds. We have also added a detailed explanation in Figure 28 to construct the alternating speed conditions.
  6. We have adjusted the images to ensure they are of high quality.

We hope that these modifications address your concerns and improve the quality of our manuscript. Thank you again for your valuable feedback and your time in reviewing our work.

Sincerely, Le Li

Round 2

Reviewer 4 Report

Authors have addressed my comments. I believe that the paper is now suitable for publication.